# Challenges and Perspectives for Integrating Quinoa into the Agri-Food System

**DOI:** 10.3390/plants12193361

**Published:** 2023-09-22

**Authors:** Irfan Afzal, Muhammad Zia Ul Haq, Shahbaz Ahmed, Abdelaziz Hirich, Didier Bazile

**Affiliations:** 1Department of Agronomy, University of Agriculture, Faisalabad 38040, Pakistan; zia48001@gmail.com; 2Department of Crop and Soil Sciences, Washington State University, Pullman, WA 99164, USA; shahbaz.ahmed@wsu.edu; 3African Sustainable Agriculture Research Institute, Mohammed VI Polytechnic University, Laayoune 70000, Morocco; abdelaziz.hirich@um6p.ma; 4CIRAD, SENS, F-34398 Montpellier, France; 5SENS, CIRAD, IRD, University Paul Valery Montpellier 3, University Montpellier, 34090 Montpellier, France

**Keywords:** quinoa, biodiversity, climate resilience, food security, nutritional security, sustainability

## Abstract

Quinoa is a highly nutritious and abiotic stress-tolerant crop that can be used to ensure food security for the rapidly growing world population under changing climate conditions. Various experiments, based on morphology, phenology, physiology, and yield-related attributes, are being conducted across the globe to check its adoptability under stressful environmental conditions. High weed infestation, early stand establishment, photoperiod sensitivity, loss of seed viability after harvest, and heat stress during its reproductive stage are major constraints to its cultivation. The presence of saponin on its outer surface is also a significant restriction to its local consumption. Scientists are using modern breeding programs, such as participatory approaches, to understand and define breeding goals to promote quinoa adaptation under marginalized conditions. Despite its rich nutritional value, there is still a need to create awareness among people and industries about its nutritional profile and potential for revenue generation. In the future, the breeding of the sweet and larger-grain quinoa varietals will be an option for avoiding the cleaning of saponins, but with the risk of having more pests in the field. There is also a need to focus on mechanized farming systems for the cultivation, harvesting, and processing of quinoa to facilitate and expand its cultivation and consumption across the globe, considering its high genetic diversity.

## 1. Quinoa’s Superiority over Other Cereals

There is a high amount of pressure on our food system to feed the rapidly growing world population and promote the sustainability of different resources related to agriculture and environmental protection. Intensive agriculture, especially the use of higher inputs over the last few decades, imposes a severe threat to the sustainability of our agro-ecosystem [1].

Modern breeding and biotechnology have vastly improved the production of major crops over the past few years; however, an overdependence on conventional crops has resulted in reduced crop diversity in the fields [2]. Moreover, global warming is also threatening food security by decreasing the yield of important grain crops due to the rise in temperature [3]. Hence, the search for alternative crops has become critical not only to improve the nutritional status of foods, but also to guard against climate change. This includes a need to introduce crops that are resilient to climate change in order to feed the world’s growing population. Over the past few years, food crops with high genetic diversity, the ability to tolerate abiotic stresses (drought, salinity, high temperature, and frost), higher profit margins, and higher nutritional profiles and values have gained attention [4,5]. One example of a genetically diverse and highly nutritious crop is quinoa, which has gained a huge amount of attention from the world [6]. Beyond its area of origin, it is now being cultivated in 120 countries around the world [7]. It is a dicotyledonous, annual, and self-pollinated plant, which has been grown in the arid and Altiplano areas of South America for centuries. It has an excellent potential to be adopted in a wide range of altitudes, ranging from 0 to 4000 m, with an ability to grow and produce seeds in warm environments and in extremely cold temperatures [8]. Quinoa is a pseudo-cereal with a high nutritional profile and a great ability to survive in saline soils [9]. It has low water requirements as compared to traditional cereal crops (Table 1).

There has been a remarkable increase in the cultivated area of quinoa in the last few years (2000–2019), especially in the Bolivian region, with increases from 35,690 to 64,789 ha^−1^, and in Peru, with increases from 27,578 to 37,625 ha^−1^. The major importers of their harvests are the United States of America (53%), Canada (15%), France (8%), Germany (4%), the Netherlands (4%), Australia (3%), and the UK (2%) [10]. A crop like quinoa, which has a great potential to survive against stresses, is an ideal option to ensure food security, decrease pressure on conventional crops, and increase farm productivity [11]. The quinoa plant has a great potential to minimize hunger by directly enhancing productivity under marginal environmental conditions where our main conventional crops have failed to perform [12]. It is the need of the hour to integrate quinoa into the agri-food systems for food and nutrient security because of its resilience to climate change.

### 1.1. Extraordinary Nutritional Properties

Quinoa is a rich source of nutrients, has a positive impact on health, and plays a significant role in reducing various diseases. Due to its high nutritional value, quinoa can be used as an alternative to our conventional food crops (wheat, maize, and rice), and its flour can be mixed with cereal grain flours to improve the nutritional status of conventional food crops. Due to its great genetic variability, the starch content in quinoa seeds varies approximately from 52% to 74% (dm). The starch properties determine the quality of the quinoa seed. The presence of all the essential amino acids determines the quality of a protein, which directly impacts the nutritional profile of a food. Protein contents in quinoa seeds vary from 9.1 to 15.6% [13], and it lacks gluten, which makes it ideal for gluten-allergic people. Quinoa has an excellent essential amino acid proportion, with a high percentage of lysine (5.1 to 6.4%) and methionine (0.4 to 1%) [14]. Due to it being well-balanced and having of all the essential amino acids, quinoa grain protein is considered superior to other cereals. Its leaves can be used for human consumption but also as protein-rich animal fodder [15].

The average oil contents of quinoa grains range from 2 to 9.5%, and mainly comprises linolenic acid (omega 3), which is helpful against cardiovascular diseases. It also enhances insulin sensitivity. The concentration of oleic acid (omega 9) and linolenic acid in quinoa are 27.7% and 38.9%, respectively. There is a region-to-region variability in the lipid content of quinoa. The Andean-region genotypes contain more lipid contents, primarily linolenic acid (4.8%) (omega 3) and linoleic acid 50.2% (omega 6) [16]. A considerable amount of micronutrients, especially iron, copper, calcium, magnesium and zinc, are present in quinoa seed [17]. Quinoa seed also contains a relatively higher amount of vitamin E, B2, and carotene than our conventional cereals [18].

**Table 1 plants-12-03361-t001:** Nutritional profile, global production, genome description, and abiotic stress tolerance of quinoa in comparison to the main cereals.

	Quinoa	Wheat	Maize	Rice	Publication
Nutritional profile					
Crude protein (% dry weight)	12–20	12	8.7	7.3	[19,20,21,22]
Total fat (% dry weight)	5	1.6	3.9	0.4	
Fiber (% dry weight)	5–10	2.7	1.7	0.4	
Total Carbohydrates (% dry weight)	59.7	70	70.9	80.4	
Gluten presence	Gluten free	12–14%	Gluten free	Gluten free	
Glycemic index	53	43	66	56	
**Minerals (mg/100 g dry weight)**					
Magnesium	249.6	169.4	137.1	73.5	[23,24]
Calcium	148.7	50.3	17.1	6.9	
Iron	13.2	3.8	2.1	0.7	
Potassium	926.7	578.3	377.1	118.3	
Phosphorus	383.7	467.7	292.6	137.8	
**Vitamins**					
Niacin	0.5–0.7	5.5	1.8	1.9	[25]
Thiamine	0.2–0.4	0.45–0.49	0.42	0.06	
Folic Acid	0.08	0.08	0.03	0.02	
Riboflavin	0.2–0.3	0.17	0.1	0.06	
**Global production perspective**					
Global market price (USD Tons^−1^)	3580	205.76	143.91	376.00	[26,27,28]
Grain yield (tons ha^−1^)	0.76	3.49	5.87	4.76	
Global production (million tons)	147	770	1210	787	
Global cultivated area (million ha)	0.191	220.75	205.87	165.25	
**Genome organization**					
Ploidy level	Allotetraploid	Tetraploid	Diploid	Diploid	[29,30]
Genome size	1.5 Gb	~17 Gb	2.4 Gb	2.4 Gb	
Chromosome no.	36	42	20	24	
Genes annotated	62,512	3685	330	56,284	
**Abiotic stress tolerance**					
Salinity stress	150–750 mM NaCl	125 mM NaCl	ModeratelySensitive to salt stress	4–8 dSm^−1^	[31,32,33,34,35,36,37,38,39]
Heat stress	35 °C	32 °C	36 °C	40–45 °C	
Drought stress (water requirement)	300–400 mm	325–450 mm	500–800 mm	450–700 mm	

### 1.2. Resistance to Adverse Environmental Conditions

Almost 50% of agricultural productivity is lost due to a wide range of abiotic stresses, i.e., salinity, drought, heavy metals, waterlogging, frost, excess heat, and UV-B. Most of these stressors usually occur in combination [40]. Quinoa, as it is drought-tolerant, has excellent potential for adaptation to the extreme arid conditions of Northern Chile and Argentina, Peru, and Bolivia [41]. Its cultivation has been expanded to the arid and semi-arid regions of Asia, the Mediterranean, North Africa, and the Near East [42]. The mechanisms that quinoa usually adopts against drought stress are classified into three categories. The physiological strategies include plasma membrane stabilization, stomatal conductance, antioxidant defense, plant growth regulation, and osmotic adjustment. The avoidance mechanisms include deep root systems and molecular approaches. Anthesis and milking are the two most drought-sensitive stages for quinoa [43]. Osmolyte accumulation, the synthesis of ROS, the accumulation of soluble sugars and proline are other mechanisms in quinoa that are involved in the adjustment of cell osmotic potential. These mechanisms enable the quinoa plant to survive and produce seeds under drought conditions [44]. 

Salt stress is another obstacle to sustainable crop production. Plants grown in saline soils show impaired growth due to a salt-induced osmotic effect, nutrient imbalance, specific ionic effect, oxidative damage due to higher levels of reactive oxidative species (ROS), and an alteration in the endogenous level of hormones [45]. Quinoa has a great potential to grow under adverse climatic and edaphic conditions [46]. Some specific quinoa accessions perform well even under sea water [47,48]. It is also a well-known facultative halophyte, due to its ability to maintain osmotic potential in its lower leaves. Because salt water is not a physical requirement for growth, it can perform well under canal water [49]. Quinoa crops have the potential to play a remarkable role in maximizing productivity and farmers’ incomes in the arid regions of the world due to its potential defenses against adverse environmental conditions [50]. Under saline conditions, the quinoa plant produces companion solutes, e.g., soluble sugar, prolines, and glycinebetaine [51], and increased antioxidants and K/Na [52]. Glycinebetaine is a betaine derivative, a major osmolyte in quinoa which makes it capable of tolerating adverse ecological stress situations [53]. 

Frost is considered one of the main obstacles limiting agricultural productivity, especially in the high Andean regions. Quinoa is less affected by frost than all the other crops grown in this region due to its ability to tolerate frost using its specific mechanisms. Various studies conducted in greenhouses and phytotrons have shown that cultivars from the highlands of Peru (Altiplano), usually cultivated up to 3800 m above sea level, have the potential to tolerate low temperatures down to −8 °C for 4 h, while the other cultivars from the Andean region tolerate the same temperatures for 2 h [54]. The quinoa plant can survive in the extremely low temperatures of down to −4 °C in the southern region of Bolivia in South America, and successfully grow at an altitude of 3.5 to 4.1 km above sea level. It has the capability to survive at freezing temperatures [55]. Some experimental trials conducted at low temperatures have shown that its vegetative growth is promoted even at −16 °C. Blanket-type isolations are formed in quinoa leaves and buds that enhance its resistance against frost [56].

### 1.3. Adaptability to Agro-Ecological Extremes

The quinoa plant has a great potential to survive under a wide range of stressful environmental conditions. It can tolerate huge ranges of fluctuations in temperature [18]. Hence, it can be successfully grown in the Himalayas and in Africa. In 1999, quinoa was introduced into the diverse agro-climatic conditions of Morocco [7]. Quinoa was introduced in Africa, North America, Asia, and Europe during the 20th century [26,57]. In the 1980s, England, Denmark, and the Netherlands were the first European countries which started research on it. In Asia, China is one of the leading countries in quinoa production and industrialization. In 2019, quinoa’s area was increased to 16,670 hectares in 25 provinces of China from 12,000 ha in 2018 [58], and currently 18 varieties are registered in China. The first quinoa variety in Pakistan was registered in 2019, while quinoa was introduced there in 2009 [59]. But, even having more than 6000 landraces in the Andes, less than 60 varieties are registered in national lists and catalogues today, which would permit the crop’s cultivation in new countries [8].

The ability of quinoa plants to synthesize protein-rich grains under diversified environmental conditions makes it an important and economically viable crop to grow in a wide range of regions [18]. In Kenya, quinoa research results showed a high grain quality and a greater yield as compared to the Andean region of South America. These results show that quinoa has a great potential for adaptation under diverse and different environmental conditions [60]. 

In food-deficient countries, especially in Africa, quinoa, as a climate-resilient and high-nutritional crop, could contribute to the reduction of poverty and enhance the food supply of this region. In Colombia and in Kenya, some quinoa genotypes can yield 4 t ha^−1^. The high-yielding potential of quinoa makes it an important crop for this region to ensure food security in the future [61]. Recently, the yield potential of seven quinoa genotypes was explored in the hot–arid regions of North Africa, and it was found that some genotypes are high-yielding and have better grain quality with short harvesting maturity [62]. Despite the wider adaptability of quinoa, its yield is quite varied, from 108 kg ha^−1^ to 9667 kg ha^−1^ around the world [35].

## 2. Challenges

### 2.1. High Weeds Infestation

Weeds infestation is the main constraint to quinoa production in low lands and temperate climates [63]. Significant damage in the form of yield loss has been reported due to high weeds infestation. The severe problems due to a high weeds population occurs from the early crop establishment stage to the visible floral bud initiation stage. Redroot pigweed (*Amaranthus retroflexus* L.), barnyard grass (*Echinochloa crus-galli* L.), common purslane (*Portulaca oleracea* L.), bermudagrass (*Cynodon dactylon* L.), common lambs quarters (*Chenopodium album* L.), and purple nutsedge (*Cyperus rotundus* L.) are the most common weeds observed in quinoa fields [64]. Among all these weeds, redroot pigweed is the most dominant in quinoa fields and difficult to control due to the same botanical family which hinders selective weed management [63]. In Europe, with cereals in the rotation, common lambs quarters is the main weed and it is very difficult to control. 

Despite the increase in weed management over the past few years, there are limited data that have been reported regarding agronomic practices, especially weed management practices, for quinoa crops. Manual hand hoeing is the most commonly and widely adopted practice to control weeds in quinoa fields during early crop establishment [35]. Manual hoeing is very expensive, costing up to USD 1000 per acre in China, and thus increases the cost of production of this crop. Despite heavy losses in yield due to weeds, there is no registered herbicide for weed removal in quinoa fields, and the international demand for quinoa is oriented to organic agriculture that bans all of them. Initial studies have also observed that the use of some types of herbicides, especially imazaquin, causes phototoxicity in young seedlings [65]. The evidence about the allelopathic effects of quinoa that has been reported could be used in cases of integrated weed management [66]. The emergence of weeds, their growth, and competition with quinoa crops are severely dependent upon the tillage system and nitrogen application [67]. More recently, Langeroodi et al. [63] reported that an integrated approach of adopting rye as a cover crop along with chemical weed control under minimum tillage has the potential to reduce weed density in quinoa cultivation. Furthermore, metribuzin, pursuit, and pendimethalin were suggested as effective herbicides for the control of weeds in quality fields.

### 2.2. Disease and Insect Pest Attack

The damage to a plant caused by disease and pest attacks can severely reduce the productivity of a crop [Table 2]. According to an estimate, approximately 0.2–0.3% of crop yield is reduced annually even in those fields where genetically disease- and insect pest-resistant cultivars and pesticides are applied. New and emerging diseases affecting quinoa have been reported due to its wide cultivation, from its Andean area of origin to new environments with high temperatures, weather fluctuations, and intense precipitation around the world [68]. Quinoa crops are exposed to attack from a wide range of microbes, with various intensity levels depending on the different environmental conditions. 

Downy mildew is a major fungal disease which infects quinoa around the world. Its contribution to the yield loss is approximately 33–58%, even in the resistant cultivars [69]. Under favorable climatic conditions, yield losses may reach 100%. Mildew in quinoa crops is caused by oomycete *Peronospora variabilis* [70]. It attacks the plant foliage and causes a yellowing or reddening of the plant leaves. Depending on the different genotypes, severe damage can cause up to 100% defoliation. This fungal disease requires specific environmental conditions for proper germination and infection. Humidity above 80% and temperatures from 15 to 20 °C are the optimum conditions for its proliferation [69]. It is widely spread because of contaminated seeds which have the pathogen [69]. Moreover, *C. album* acts as an alternative host for *P. variabilis*, which ultimately infects quinoa in fields [71]. 

Damping-off is another disease that was first reported in quinoa plants in 1980. *Sclerotium rolfsii*, a causal organism, was separated from the infected seedlings [72]. The fungus *Fusarium*, along with *Rhizoctonia solani*, were also isolated from quinoa fields at the International Potato Center in Peru [69]. A study showed that *Pythium aphanidermatum* and *Fusarium avenaceum* were the causal agents of the damping-off disease of quinoa seedlings [73,74]. From germination to the end of the cotyledon leaves, the quinoa plant is highly susceptible to this disease. Its susceptibility to *F. avenaceum* is at its peak from the emergence of cotyledon leaves to the first pair of true leaves stage.

Brown stalk rot was first reported in Puno (Peru) in 1974–75 and has been frequently observed in the Peruvian Altiplano and in various quinoa-growing regions of North America and the UK. The presence of wounds, low temperatures, and high relative humidity are the optimum conditions for the spreading of this disease [75]. A dark brown lesion of 5–15 cm in length appears on the stem and inflorescence portion. Shrunken stems, defoliation, and chlorosis are other symptoms of this disease [72].

Pest attack damage in quinoa ranges from reduced yield to the death of the plant. The major pests in quinoa fields include borers and leaf miners, which eat the leaves, stems, roots, and grains. On the foliage, chewing and sucking insects and stem cutters are dominant. Birds, rodents, and several insects also attack mature quinoa grains, causing significant damage to the crop [60]. Recently, Cruces et al. [76] indicated higher pest pressure on quinoa grown at lower elevations than in highlands. 

Quinoa insects are classified as defoliating and biting insects, mining and grain-destroying, and cutworms. Most insect pest attacks are observed in quinoa during the vegetative and reproductive growth stages, and under storage conditions [77]. However, the yield loss caused by the different insects varies according to the region. The major insect pests of quinoa are the moths *Eurysacca melanocampta* (Meyrick) and *Eurysacca quinoae* Povolný (Lepidoptera: Gelechiidae), while thrips and aphids are considered of minor importance [76]. Aphid (*Aphis gossypii*) attacks on quinoa plants were also observed in Egyptian regions, which caused significant damage to the crop. Aphids suck the plant sap and their feeding on the leaves causes the distortion and curling of leaves, which ultimately reduces the plant’s photosynthetic ability [78].

**Table 2 plants-12-03361-t002:** Diseases and insect pests of quinoa.

Disease/Insect Pest	Reporting Area	Effects	Growth Stage	Management	Publication
Western flower thrips (*Frankliniella occidentalis*, *N. capsiformis*)	Peru	Discoloration, distortion, premature drying, and shedding of leaves, flowers, and buds.	Late crop stages.	Methomyl + emamectin benzoate	[76]
Serpentine leaf minor (*Liriomyza huidobrensis*)	Italy	Premature leaf drop.	Middle crop maturation stages.	Methomyl + dimethoate	[76]
Potato aphid/aphid complex (*Liriomyza huidobrensis, R. rufiabdominale, Myzus* sp., and *Macrosiphum* sp.)	Italy, Peru	The development of sooty mold on the leaves.	60 days after sowing.	Methomyl + dimethoate	[76]
Hemipteran Pests (*L. hyalinus*, *N. simulans*)	Italy, Peru	Seed- or leaf-feeding insects.	Grain-filling stage.	Dimethoate and methomyl	[79]
The noctuid complex (Helicoverpa, Copitarsia, Copitarsia, and Agrotis genera)	Chile, Argentina, Ecuador, and Colombia	Adult insects feed only on flowers’ nectar and other sweet secretions. Major damage has been observed during the flowering and physiological maturity stages. Because these pests enter into the panicle rachis leading it to break off, resulting in defoliation. Noctuid species also feed on developing grains.	The flowering and dough stages are the most sensitive stages for these insect attacks.	1. Crop rotation. 2. Using light traps. 3. Using pheromones traps. 4. Preventive treatment (use of lime sulfur which effects the insects’ central nervous system). 5. Use of insecticide (spinosad).	
Quinoa moth complex (*Eurysacca melanocampta*, *E. quinoae*, and *E. media*)	Argentina, Chile, Colombia, Bolivia, and Peru	Photosynthetic area is reduced and first-generation larvae feed on the leaves’ parenchyma, roll leaves, and tender shoots, and destroy the developing inflorescences. Second-generation larvae damage the developed inflorescences, milk and dough stage grains, and mature grains, and ultimately cause a 15–60% reduction in yield.	Grain development and physiological maturity.	Use of spinosad as an eco-friendly insecticide.	[80,81]
**Quinoa crop diseases**					
Downy mildew	Argentina, Colombia, Bolivia, Ecuador, Chile, Peru, Canada, Mexico, USA, Portugal, the Netherlands, France, UK, Sweden, Denmark, Italy, Kenya, and India	Primary effect of this disease is on the leaves but symptoms can also appear on the stems, inflorescences, branches, and on the grains. Initial symptoms appear on the leaves as small, irregular spots that may be chlorotic, yellow, grey, pink, orange, or red, depending on the plant color.	The initial developmental stages are mostly affected by this disease. Optimal conditions for downy mildew development are relative humidity (>80%) and temperatures of 18–22 °C.	1. Genetic resistance. 2. Use of quality seed. 3. Use of eco-friendly fungicides (liquid extracts of horsetail and garlic). 4. Use of fungicide (metalaxyl).	[82]
Brown stalk rot	Peru, North America, and UK	A dark-brown lesion of 5–15 cm length appears on the stem and inflorescence portion. A shrunken stem, defoliation, and chlorosis are some other symptoms that may occur with this disease. Pathogens are mostly located in the stem and inflorescence.	Mostly occurs in the early developmental stages.	1. Spray with carbendazim. 2. Mancozeb solution (70% mancozeb diluted to 1000 times).	[83]
Root rot	South American regions	The major symptom of this disease is black rot on the quinoa root. This causes the very low supply of water and nutrients to the roots and results in yellowing and ultimately death.	It is a soil-borne disease.	1. Use of hymexazol (50% content is diluted 1200–1500 times). 2. Soaking of quinoa seed in thiram for ten hours before sowing.	[83]
Leaf spot	Major quinoa disease	High temperature favors this disease. Initially there is the formation of light spots on the leaves’ surface; later the leaves dry out and fall off.	Seed-borne disease.	Use of diniconazole powder (12.5% diniconazole is 30–40 g per 667 m^2^ for spray).	[47]
Gray mold	Cambridge	The stem and inflorescence of quinoa are mostly effected by this disease.	Stem elongation and panicle formation are the most sensitive stages.	Spray of iprodione diluted 1000–1500 times.	[83]
Quinoa Diamond Black Stem	Puno, Peru	Ascochyta leaf spot and stem necroses.	Stem-specific fungal agents.	Mancozeb and azoxystrobin fungicides.	[70,84]
Sclerotium	Cuzco, Peru	Whitish-to-grey stem lesions and sometimes conjugated ones.	Infected seeds and soil and crop debris.	Methyl benzimidazole carbamates and dicarboxamides.	[70]
Damping-off	Southern California, Nihon (Japan), and Peru	High moisture causes lesions on the quinoa leaves. High soil moisture causes the formation of diseased seedlings and wilting.	Seed- and soil- borne disease.	Phenyl-pyrroles (P.P. fungicides) and dimethylation inhibitors (DMIs).	[70]
Viral diseases	Peru, Bolivia	Chlorotic local lesions and severe systemic mosaic, leaf deformation, wilting, stunning and, finally, the collapse of the plants.	Seed-borne virus.	Seed sanitation.	[70,84]

### 2.3. Stand Establishment

Stand establishment is the most critical and sensitive phenological stage in quinoa. Good stand establishment contributes towards high yield and seed quality. A poor germination percentage and low crop stand establishment are the major issues faced by the farmers growing quinoa in saline or various other marginal lands. A number of research trials conducted by the FAO in the UAE showed high variations in terms of stand establishment [28]. Temperature and moisture contents have a significant effect on quinoa seed germination and ultimately on early stand establishment. Quinoa seed germination and seedling development are highly influenced by environmental conditions and severely affected by temperature [85]. The maximum germination was observed at temperatures of 20–35 °C, while a significant reduction was observed in germination at temperatures below 15 °C and above 45 °C. The number of aborted seeds sharply increased with an increase of temperature above 35 °C, and a total loss from 50 °C [86,87]. Moreover, low temperatures inhibit quinoa seed germination due to embryo death [88]. 

Quinoa germination is still challenging for researchers [89], as its stand establishment is very tricky due to the small-sized seeds and, in particular, the low soil moisture content that significantly reduces its emergence in the field [90]. Due to its orthodox behavior, quinoa seed has the ability to dehydrate after drying up to 5% of its moisture contents without losing its viability. These types of cells are dehydrated from the loss of vacuolar water from the mother plant to the seed during the maturation process, which helps to maintain seed viability and storage potential [91]. Conservation approaches to orthodox seeds require negligible physiological activity. But some non-enzymatic reactions take place when moisture contents are low, which leads the seed to age, bear alterations on its functional proteins which weakens the seed’s metabolic system, and reduces its ability to defend and repair itself from free radical damage up to the end of the germination process [92].

### 2.4. Saponin-Free Quinoa

Saponin is a bitter chemical substance present on the outer surface of the seed. It is used to make foamy products like toothpastes, shampoo, shaving creams, soaps, etc. This bitter substance is a big restriction on the direct consumption of quinoa seed for human food [93]. The total seed saponin content of quinoa can be up to 1.56% (1.56 g/100 g), which is lower than that of other plant parts except the leaves, which have 0.97% (0.97 g/100 g). Quinoa roots contain the highest amount of total saponins contents (13.39 g/100 g) [13,94]. About 67.5% of the total saponin in the grain is present in the pericarp, while the rest is present in the endosperm and the other internal layers of the quinoa seed [95]. Some abiotic factors, such as salinity and drought (or low irrigation), affect the total saponin contents, respectively, positively and negatively [13,96]. However, it was observed that the growth of fungi are restricted in high saponin-containing quinoa grains [97]. In a recent study, it was found that the shortest harvest maturity quinoa genotype had lower grain saponin contents (0.62 g/100 g DM) than the long-duration quinoa genotypes, which have high grain saponin contents (1.92 g/100 g DM) [62]. 

The saponin removal process is the most critical operation in quinoa seed processing. Manual, artisanal, and industrialized technical methods are used to eliminate saponin from quinoa seeds, while washing is the most widely used method in the world for large volumes with commercial destinations [Table 3]. Physical abrasion combined with water cleaning can cause important losses of brut production during post-harvest operations (up to 20% in some local farming contexts). The production of quinoa without saponin has been a breeding target [98] to save this cost of its elimination during processing, during which some important nutrients are also lost. Hence, it would be interesting to introduce some sweet varieties of quinoa to enhance its edible consumption and ensure market sustainability. The Kancolla variety from Peru is a famous sweet variety of quinoa consumed in the same ways as wheat (*Triticum aestivum*) and rice (*Oryza sativa*). It is a pseudo-cereal rediscovered by crop science researchers from industrialized societies and was selected for its high potential of tolerance to extreme temperatures and resistance to various diseases [99]. Improved methods for the removal of saponin contents without any modification to its nutritional value are encouraged for maintaining the specific nutritional traits of the bitter varieties. Sweet genotype selection with low seed saponin contents, bold grain size, and high seed yield are the major breeding goals. Even if this appears as the main objective for many breeders all over the world because it can simplify some of the operations in the quinoa value chain, we do not forget the natural protection conferred by saponins to quinoa plants in the field and during seed storage. Sweet quinoa genotypes should be selected as early as possible during crop development to boost the selection process. Further research is also needed to search markers for the indirect selection of sweet genotypes [100].

### 2.5. Seed Longevity after Harvest

Despite its great potential for growing under adverse environmental conditions, quinoa seed quality is adversely affected due to low germination percentage and vigor [106]. Quinoa seed loses its viability very rapidly compared to our conventional cereals due to integument porosity. This allows quinoa seed to gain and lose moisture contents very rapidly, which can initiate germination even in the panicle [107]. Quinoa seed shows an orthodox behavior because of its ability to dehydrate up to equilibrium the water contents of its environment, which enhance its ability against drought by up to 5% without losing its viability [108]. The vacuolar water loss from the seed cells supplied by the mother plant to the seed during the maturation process results in the cells being dehydrated, which maintains seed viability and storage potential [91]. Minimum physiological activities are required for conservation techniques for orthodox seeds. Seed moisture content is a major factor affecting the damaging rate and aging reactions during seed storage [92]. The optimum condition for storing quinoa seed is at about 10% moisture content to enhance its longevity. Under unsuitable storage conditions, with temperatures of 10–20 °C and a relative humidity of 75–80%, quinoa seed loses its viability rapidly in a very short period of time [109]. Storage conditions significantly influence the dormancy pattern output and seeds’ longevity, along with the genotypes for contrasting environmental conditions [92,110]. A recent study elucidated that an increase in storage duration along with temperature contributed to significant changes in the grain moisture, and the nutritional and color properties of quinoa [111].

### 2.6. Photoperiod Sensitivity

Quinoa has been successfully grown in South American regions, primarily by Peruvian and Bolivian peasants from ancient times. However, during the last few decades, quinoa cultivation has been expanded worldwide [112,113,114]. Quinoa has an excellent potential for adoptability under marginal and stressful environmental conditions. Furthermore, quinoa plants can stimulate successful growth and seed production even at high-temperature ranges, depending upon the genotypes and location [115,116,117,118]. But pollen viability is adversely affected at a temperature above 40 °C [118].

The photoperiod is the duration of lightness, with alternating darkness, in the daily cycle of 24 h. As we move toward the poles, the light and dark duration difference is more extreme; at the equator, the photoperiod is constant with equal light and dark hours/day. The earth’s annual rotation causes a significant change in the hemisphere photoperiod throughout the entire year. The main obstacle to the adaptability of quinoa in the northern hemisphere is the photoperiod. Quinoa is a facultative short-day and day-neutral plant [119,120]. An experimental trial conducted in Demark showed that flowering induction using different photoperiods was not the most important problem for quinoa adaptability in the northern hemisphere. However, a quinoa plant’s exposure to a longer day length delays the maturation stage. The plant height and biomass of quinoa significantly increase in response to a longer day length [120]. Seed development deteriorates the crop yield of photoperiod-sensitive cultivars due to more stem and lower-leaf growth [121]. The leaf color of some quinoa genotypes turned red in response to a change in the photoperiod [122,123], but in many other cases the original color disappeared. A long day length (16 h) increased the seed yield of all quinoa varieties except that of the short-day plant [124]. A quinoa plant’s growth cycle, morphological appearance, and seed quality are influenced by a constant day length of 16 h and 8 h, if the plants are short-day or neutral.

### 2.7. Stem Resistance

Stem lodging is a serious issue in some quinoa genotypes, which causes a significant loss in seed yield. This issue is genotype-dependent and most common in sandy soils, which have a lower anchorage capacity than loamy and sandy soils. Major advances in quinoa seed yield can be accomplished by introducing and developing genotypes with high resistance to stem lodging, and growing them in a productive environment managed using appropriate farming techniques. Resistance to stem lodging ensures proper seed filling to ensure minimum harvest losses. Many individual characters affect resistance to stem lodging, including the diameter of the stem, plant height, the stem outer-wall thickness, and the type of root system. The genetic control of the stalk strength is quantitative [123].

### 2.8. Heat Stress at Reproductive Stage

Despite its high potential for tolerance against various stresses, high temperature is one of the most vital abiotic stresses during plant growth, and can cause a significant reduction in yield in combination with drought [125]. Heat stress induces morphological changes, like the inhibition of root and shoot growth, enhanced stem branching, and anatomical changes with reduced cell size and enhanced stomatal and trichome densities [40]. In addition to the morphological and anatomical changes, the physiological consequences of heat stress include increased membrane fluidity; cytoskeleton instability; protein denaturation; variations in respiration, photosynthesis, and carbon metabolism enzymes activity; and changes in phytohormones, including ethylene, ABA, and salicylic acid [126]. 

Quinoa has a great potential to tolerate a wide range of temperatures (−8 °C to 35 °C) and relative humidity (40% to 88%), but that is highly dependent of the genotype and developmental stage [54]. A sudden rise in temperature during flowering and seed setting can cause a significant reduction in the yield, and is one of the primary restrictions to the global expansion of quinoa. For example, studies in Italy, Morocco, Germany, Portugal, India, Egypt, Mauritania, and the United States have reported that high temperatures reduced quinoa seed yield. Temperature above 35 °C near Pullman caused a significant reduction in the seed yield due to empty seeds or seeds lacking in inflorescence. Similarly, Bonifacio [127] observed both the reabsorption of quinoa seed endosperm and the inhibition of anther dehiscence in its flowers due to high temperatures (35 °C) at the anthesis stage. Various experimental results have shown that night temperatures between 20 and 22 °C (~4 °C above the night ambient air temperature) have a negative effect on plant biomass, seed number, and ultimately, on the yield; however, the protein contents and harvest index were not affected. Another experiment conducted by Hirich et al. [7] in the UAE which tested the performance of several quinoa varieties under four different temperatures—24, 29, 35, and 42 °C—showed that temperatures above 30 °C negatively affect quinoa growth and productivity by inhibiting photosynthetic activity and reducing the flowering rate and grain filling.

### 2.9. Agronomic and Socio-Economic Constraints to Its Cultivation

Recently, quinoa cultivation has spread to various regions of the world because of its high nutritional properties. It is considered as a superfood because the seeds contain all the essential amino acids, minerals, vitamins, and trace elements. In the past four decades, quinoa production has increased 252% and 612%, respectively, in Peru and Bolivia. Traditionally, quinoa was cultivated using crop rotation. But nowadays, a rapid increase in quinoa seed demand, mainly exported from the Andes, has shifted farmers towards more and more intensive monoculture, with a reduction in the fallow period and the use of heavy agricultural machinery, principally for sowing, which has resulted in a considerable increase in erosion and soil degradation, with various other side effects on social organizations [128]. The use of heavy agricultural machinery has enhanced the populations of pests in the subsoil, demanding pest control management [129]. To ensure continuous quinoa production, farmers must leave fallow periods ranging from 6 to 8 years to control nutrient depletion and soil erosion [130]. The monoculture of the quinoa crop in the Andes is specific to the Southern Altiplano of Bolivia, with altitude desert conditions that preclude any other crops. However, the recent intensification of demand for quinoa favors the use of a reduced number of genotypes, which causes a significant reduction in crops genetic diversity, and could increase its vulnerability to various biotic and abiotic stresses [131]. At least 20 quinoa commercial varieties presently exist in the Peru region; however, about 90% of the entire quinoa produced for export are covered by four quinoa genotypes [132]. The intensification process could be threatened by the Andean agro-ecological conditions, with its sandy and volcanic soil, which is characterized by high salinity levels, low moisture-retention capacity, and low organic matter [133].

It was shown in a comprehensive study conducted in Morocco, which analyzed the local value chain, that there are many agronomic and economic barriers limiting quinoa development and expansion [7]. Regarding the agronomic constraints, the farmers claimed that there is a continuous loss of quinoa productivity after successive harvests due to the lack of high-quality and genetically stable seeds selected for the next agricultural campaign. In fact, farmers always use the harvested seed to sow in the next season, which leads to more genetic segregation and, thus, yield variability and instability; this is also accentuated by bad storage conditions that affects seed germination. In addition to the lack of high-quality seeds, the farmers stated other agronomic constraints, such as phytosanitary problems (caterpillar attack, downy mildew, bird attacks, weeds, etc.) and high production costs, which are mainly associated with the manual production mode. One of the big constraints limiting quinoa’s value chain development and upscaling is the low local market demand and the lack of access to international market channels. Therefore, it is recommended to deploy much effort to quinoa promotion and raising awareness among consumers [134]. Another important point is to involve farmers and consumers from the beginning, in a pathway that considers local needs and tries to give them a specific solution. The example of Bhutan, with the development of new quinoa-based dishes, was a key element of quinoa’s success story in the country [135]. 

Production enhancement also raises concerns regarding sustainability problems in cultivation areas [136,137]. Quinoa is very susceptible to market vagaries, such as rising prices which enhance competition. A lack of awareness among people about the nutritional profile of quinoa and the presence of saponins contents on its seed are two main obstacles to its local consumption. These cause a significant limitation in quinoa seed consumption within local diets. The prohibitive quinoa seed prices play in the favor of less nutritious food items (e.g., rice or wheat) that lack the essential micronutrients present in quinoa. Quinoa export has continuously increased since 1990, and has always been dominated by Peru and Bolivia. After the first period when domestic consumption declined, from 1961 to 1990, due to a lack of awareness about its nutritional profile, high market prices, and no effective measures to remove saponin contents [129], Andean countries have benefited from the international recognition of quinoa during the UN International Year of Quinoa in 2013. Peruvian consumption today is double of what it was in 1990 [57].

## 3. Opportunities

### 3.1. Breeding Opportunities

Experimental studies in the past few decades on breeding quinoa have made remarkable genetic improvements to productivity, pests and disease resistance, better adaptability under adverse soil and environmental conditions, and uniformity in grain quality [127]. Moreover, participatory variety selection and participatory plant breeding techniques have been successfully practiced in different regions and on different crops over the last 30 years [8]. The following breeding techniques have been used for quinoa improvement across the globe.

Different methods of conventional breeding implemented for quinoa genetic improvements include mass and individual selection, hybridization, and mutation induction. Each method has benefits and drawbacks. The organic production of quinoa in the Andes is regulated with the aid of policies set up at the expense of countrywide public and personal certifying companies; one of the demands is that the sort of methods used in this type of production should have been acquired through conventional breeding techniques. The description of these techniques is as follows.

**Mass selection** denotes the selection of a large number of advanced plants with an identical phenotypic character in their traditional or landrace genotypes. After harvesting, the seed is mixed to form new variety. This technique is applied many times in the same population to enhance the performance of the base population. The genotypes developed using this technique have a wider range of adaptability, yield stability, and an extensive genetic base for a longer period. The genotypes developed via mass selection are a combination of various populations with different phenological, morphological, agronomical, and yield-related characteristics, and responses to biotic and abiotic stresses. However, they are identical in height, maturity time, seed size, color, saponin content, and other industry- and market-preferred characters [138].

**Individual selection** is the selection of a single plant with superior attributes, in an original, genetically different plant population. The common recommendation is the selection of a plant population with the appropriate attributes to attain the improvement aims, based on space, money and time. The seeds of the selected plants are harvested separately and sown individually in the next growing period in separate rows, with adequate space to evaluate their offspring’s resistance to various diseases, height, yield, and other plant components [127].

**The hybridization technique** has played a vital role in quinoa genotypes’ improvement against biotic and abiotic stresses in the Andean region. Other experimental studies on breeding conducted outside its area of origin have described the use of hybridization technique as an important tool with which to study the genetic inheritance of important traits and to develop new quinoa genotypes highly adapted to the North American region [139,140,141]. Polyploidy, floral polymorphism, male sterility, the selection of parents, emasculation, pollination, and the management of segregating generations (F2 to F6–7) are the key factors which affect hybridization methodology [138].

**The backcross method** is used to upgrade treasured agronomic business varieties which have one or more terrible characters; these are normally related to susceptibility to various diseases or crop high qualities. To ensure favorable outcomes, the characters ought to be qualitative and dominant. An example of the backcross method is the improvement of the grain size of the genotype Patacamaya (a plant with green color and small, sweet grains), which was used as a donor parent line 1638, an accession of the Royal race type, pandela (pink), with massive and bitter grains. The backcross method has also been used for cultivated genotypes improvement with genes donated from wild varieties [127].

**Induced mutation,** as a breeding technique, is generally recommended for genetic improvement in quinoa when there is a need to change one or a few traits in the commercial and traditional genotypes used by industry and farmers. The use of suitable conventional or commercial genotypes presents a chance to directly and swiftly liberate one variety, if an ideal mutant type is observed, due to the fact that adoption with the aid of farmers and industry is ensured by the slight alteration of the basic genotype of their varieties. Conventional varieties of quinoa have valuable combinations of genes concerning their adoption to marginal environments and soils and high vitamins values. However, some have agronomically undesirable traits that lessen the yield and make their use hard in modern-day agriculture, with its high inputs and extensive regions. Among those attributes are plant heights above 2 m, extravagant branching, a long life cycle, and susceptibility to various diseases. In an identical way, a few negative fine traits can be discovered, together with high saponin content, small grains, and thick fruit layers [138].

#### Modern Biotechnology Techniques

A few techniques and equipment from modern biotechnology have been implemented in quinoa genetic improvement, which includes in vitro cellular and tissue subculture. Alternatively, DNA-based totally genetic markers and genomic assets have elevated the understanding of and capability to represent genetic diversity inside the germplasm, and to enhance traditional breeding. A few important biotechnology techniques are described below.

**Genetic shares increments and seed production:** An in vitro callus production of quinoa was carried out to evolve male sterile lines for hybrid production. The in vitro vegetative propagation of Peruvian quinoa has been performed to enhance populations of few accessions with low germination potential [142]. An agreement for the cloning of quinoa and the boom share of hybrid seeds to avoid cross-pollination was reported in Brazil.

**Double haploid breeding:** Preliminary studies were conducted on quinoa to attain doubled haploids from the in vitro cultivation of anthers (microspores) of the genotypes Blanca de Hualhuas and Rosada de Huancayo. This equipment may allow an acceleration of the development of the latest quinoa cultivars [143].

**Molecular marker development:** Molecular markers are essential in advanced plant breeding techniques. Their use has advanced conventional breeding via offering genomic gear, which is beneficial for the incorporation of genes which might be especially inspired by the way of surrounding and for others which are difficult to study, such as those that confer resistance to diseases and those that build up multiple genes for resistance to unique pathogens and pests inside the same cultivar (gene pyramidation). In addition, it has allowed plant breeders the quickest generational strength, considering that through the use of PCR a gene may be evaluated for advanced generation throughout the breeding process. It is vastly important for the identification of beneficial characters of tolerance or resistance to the principal types of abiotic and biotic stresses in the germplasm of a species, and their utility in breeding, making interspecific crosses and the choosing of germplasm with respect to those characters and preserving the core collection [7].

### 3.2. Quality Seed Production (Management Practices)

Quinoa seed quality and yield are very low under moderate agro-management practices, around 500–1000 kg ha^−1^ in the Andean countries [8,144]. With proper irrigation scheduling (310–1300 mm), plant density (10–300 plants m^−2^), and optimum nitrogen application (120–180 kg ha^−1^), seed quality and yield can be enhanced up to 1000–2000 kg ha^−1^ [8,145]. Quinoa, being a drought-tolerant plant, has a great potential to grow under low water availability. The yield and seed quality are directly correlated with irrigation [43]. Drip irrigation is a suitable method in limited water regions to enhance seed production and quality [146] (Wang et al., 2011). Quinoa seed yields can be optimized with an increasing nitrogen rate depending on the soil type and location, for example, 120 kg ha^−1^ in Germany, 310 kg ha^−1^ in Egypt, and 180 kg ha^−1^ in Denmark [140]. A remarkable reduction in seed yield, to 160 kg ha^−1^, was observed by increasing the nitrogen level [147]. These differences in seed quality and yield are due to huge genetic variability, variations in soil fertility status, crop needs, plant density, nutritional supply, and environmental constraints [148,149]. Plant density is a most important factor, which determines the yield and seed quality, and is highly dependent on the genotype, cropping strategies, and climatic conditions [150]. Plant density variations from 10 to 60 plants m^−2^ have not had significant effects on seed yields using a sprinkler irrigation system at approximately seven-day intervals. A quadratic response in seed quality and yield was observed to be related to plant density, and the maximum yield was observed at an optimum plant density of 17 plants m^−2^, with no fertilizer or water supply in eastern Austria [151].

### 3.3. Plant Genetic Resources (Seed Supply System in Developing Countries)

The number of quinoa-cultivating countries has rapidly increased, from eight in 1980 to 75 in 2014. Despite the number of established quinoa-cultivating countries, 20 other countries sowed quinoa for the first time in 2015. Due to its huge genetic diversity, it has a great potential to survive under agro-ecological extremes (temperature, soils, rainfall, and altitude), and has great tolerance against salinity, frost, and drought [142]. It can be divided into various groups or ecotypes, reflecting its dispersal from its center of origin around Lake Titicaca. Every ecotype is linked to the sub-centers of diversity and highly adaptable to its particular environments conditions [119]. The diversity of quinoa is divided into five major ecotypes [8]. Research partnerships have made the substitution of exotic quinoa germplasm easy, and have had a capable effect on its development by build-up collaborations [152]. Nowadays, quinoa is still recognized as a minor food crop with which to ensure food security and enhance agricultural productivity, and is often considered as a neglected and underutilized species with a huge potential for development under stressful environmental conditions.

Quinoa, unluckily, is not one of those species included in Annex 1 of the Treaty, which is a list of those plant species included in the multilateral system of exotic germplasm exchange among farmers, scientists, and breeders. The Declaration of Cordoba (2012), by the International Seminar “Crops for the XXI Century”, the first international action celebrating the UN IYQ 2013, proposed the addition of minor food crops to be included in Annex I to the Treaty [153]. To date, however, there has been no consensus reached. The various regulations on plant breeding and genetic resources are normally applied at antithetic levels (international, national, and local) and for various purposes (seed, genetic resources, varieties, and agricultural by-products). There is a lack of a legal framework around quinoa which would provide brief and comprehensive coverage related to all the problems associated with genetic resources and sustainable crop management [154].

The Convention on Biological Diversity (CBD) advocated a bilateral approach and benefits sharing, but its application to quinoa is difficult as the food crop is now grown internationally and is not limited to the Andean region of South America. Before 1992, twenty-five countries had established an ex situ assembling of quinoa accessions, which were distributed globally without the legal demand of a prior affiliation from the Andean countries. Several countries had set up collections earlier before the CBS came into force. Now, most of the countries have developed new varieties from exotic germplasms that are highly adoptable under their local conditions. Individuals and institutions have no permission to share quinoa germplasm apart from within their own country, as per the terms of the CBD, but this refers to only those countries that are approved by the CBD. The CBD has been approved by 196 countries (the Parties), and 168 have signed it. A few countries, in particular the USA, are not ratified by the CBD. The transfer of exotic quinoa germplasm across regions, and which these legal constraints at the global level would restrict, has contributed to part of its genetic diversity and has ensured quinoa’s adaptation under new environmental conditions to expand its cultivation globally. Quinoa germplasm exchange can be performed formally through legal provisions (Standard Material Transfer Agreements), and informally through research networks. Only 25% of the quinoa genetic material replaced is related to individual exchanges, while 75% of the exotic material is replaced via research networks [155].

### 3.4. Dry Chain Technology for Seed Preservation

Quinoa cultivation on a commercial scale requires high-quality seed, which is a major restriction for its optimum germination because of its ability to lose viability under unfavorable storage conditions. Seed deterioration during storage can be minimized by adopting dry chain technology aimed at proper drying, and then packing the seed in hermetic bags to prevent the loss or gain of moisture contents. Optimum seed germination and the maintenance of the initial moisture contents were observed when the seed was stored at an 8% initial seed moisture content in hermetic bags. A significant increase in the activity of α-amylase and the total soluble sugar, along with a remarkable reduction in electrical conductivity, the MDA contents, and reduced sugar were observed during hermetic seed storage at an 8% initial seed moisture content. Dry chain technology ensures high viability, high vigor, optimum germination, and negligible seed deterioration as compared to traditional seed storage techniques, and improves the overall seed quality, which is a major factor to enhancing agricultural productivity and ensuring food security [156]. Unfavorable storage conditions, particularly when seeds are stored at a 12 or 14% initial seed moisture content, are the main reasons for ethanol production which promotes seed deterioration and viability loss [157].

### 3.5. Seed and Grain Quality Evaluation

Because of its nutrient-rich profile and potential applications in food products, quinoa is rapidly cultivating a global market. South American regions, especially Bolivia and Peru, are considered the main quinoa producers and exporters. The United States and Canada import a major portion of quinoa seed, accounting for 53% and 15% of world quinoa imports, respectively [158]. The average yield of quinoa is 600 kg/ha, with huge variations in yield depending upon the genotype, location, and environmental factors [159].

Quinoa seed comprises many sensory attributes. Food texture explains those properties of the seed that are perceived with the tongue, finger, and teeth [160]. Quinoa has a unique texture—a little crunchy, creamy, and smooth [14]. Cooked quinoa’s texture is affected by the seed structure and its starch and protein contents. The seed attributes and structure are the major indicators affecting the textual characteristics of cooked quinoa. The major food storing tissue is the middle perisperm, which has thin walls and angular-shaped starchy grains [161]. The double-layer endospermic region contains thick-walled cells which are rich in amino acids (protein) and lipids. The embryonic region contains protein bodies and endosperm, is deficient in crystalloids, and comprises one or more globoids of phytin. The seed coat, hardness, and seed size may influence the textural properties of the cooked quinoa. Beside the physical attributes of the quinoa seed, its composition also influences the texture. Protein and starch are the major components of quinoa. The seed starch granules are very small in size (1–2 μm) as compared to rice and barley [162]. It contains very low amounts of amylose and has high amylopectin content [163], which may cause its hard texture. In addition to the grain properties and quality, the sensory characteristics of quinoa seed are also highly significant because they affect consumer acceptance and the use of the quinoa variety. The lack of a lexicon is a major restriction to accessing the sensory properties of the cooked quinoa seed. Rice is considered as a standard reference when analyzing the sensory properties of quinoa because both are cooked in a similar manner. Floral, sewer, popcorn, green beans, and sweet and sour taste are the sensory properties of quinoa seed. Consumer acceptance is of key interest to breeders, peasants, and the food industry [164,165,166]. Quinoa consumer acceptance can be affected by the demographics of the panelists, like their origin, similarity with less common grains, food culture, and assessment of a healthy diet. Sensory assessment tests are usually expensive and time consuming. Hence, the sensory panel’s correlations and instrumental data are of great interest. If correlations are present, instrumental analyses are then used to evaluate the sensory panel evaluation. 

Quinoa valorization and marketing are mainly limited by the seed quality and processing. Seeds with a high content of saponin are unlikely to be accepted and appreciated by consumers due to their bitter taste. Thus, processing operations, such as mechanical pearling, is recommended to save time and reduce processing costs. A recent study conducted by Rafik et al. [167] revealed that a pearling duration of 2 min was enough to keep the saponin content within the CODEX threshold (0.12%), and it was far better when compared to manual pearling, where the saponin content remained higher than the limit [168]. This study also indicated that pearling does not affect the seed protein or macro-nutrient content, while it does significantly reduce the micro-nutrient content.

### 3.6. Quinoa Market

The quinoa market worldwide is segmented based on its uses and applications; for instance, quinoa food products are the main market segment, in addition to cosmetics, industrial, and pharmaceutical applications. Regarding quinoa food products, their market is mainly segmented into organic and conventional quinoa seeds. In fact, as a result of the increasing population of health-conscious consumers and a rise in awareness about the consumption of organic products, the quinoa market is expanding and its demand in the global market is also increasing (Figure 1).

## 4. Conclusions and Future Trends

Quinoa, as an innovative and emerging food crop, is a rich source of protein, carbohydrates, fiber, minerals, vitamins, and other bioactive compounds. It is widely used in the synthesis of gluten-free nutrient-enriched by-products. Various experimental trials based on its morphology, phenology, physiology, and yield-related traits have been conducted around the globe to check its adoptability under different stressful environmental conditions to ensure food security for burgeoning populations [6]. It is important to develop improved varieties and promote innovative seed supply systems to support the adoption of improved varieties in the world. Over the last few decades, plant breeders have tried to develop high-yielding and stress-tolerant cultivars by using modern plant breeding techniques to enhance their adoptability under different climatic conditions for sustainable production. In the future, it will be necessary to use modern breeding programs, such participatory approaches, for a better understanding and defining of breeding goals to promote quinoa adaptation under marginalized conditions. The nutritional security of small land holders in Asia and Africa is under threat due to the changing climate scenario. Although various experimental trials have been conducted at different locations, and their findings describing the influence of abiotic stress on quinoa plants have been published, a trans-disciplinary-perspective analysis of how quinoa plants respond to a wide range of environmental conditions needs to be explored. Now, there is a need to create awareness among people and industries about its nutritional profile and the minor bioactive compounds present in the seed which can be used as raw materials for making several by-products. In the future, there will be a need to conduct research on the most suitable and inexpensive saponin-removal techniques for quinoa seed to enhance its local consumption. For the adjustment of a strategic pathway, a well-organized scientific research agenda, responsible and participatory cooperation, strong political support from national governments and international organizations, and consumer awareness is needed.

## Figures and Tables

**Figure 1 plants-12-03361-f001:**
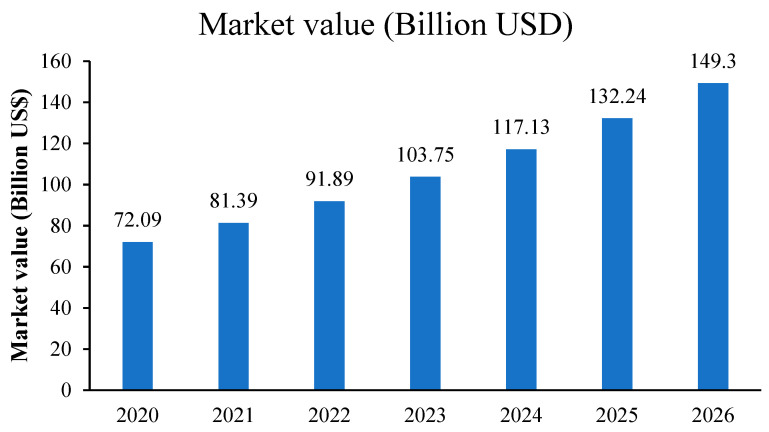
Value of quinoa market worldwide from 2020 to 2026, source: https://www.statista.com/statistics/1128506/global-quinoa-market-value-by-country/ (accessed on 8 March 2022).

**Table 3 plants-12-03361-t003:** Different methods used to measure saponin contents in quinoa seeds.

Technique Used	Seed Source/Origin	Findings	Publications
Wet washing methods. Dry methods. Genetic methods.	Not mentioned	Quinoa seed is classified as sweet if it has a foam height of 1.3 cm or less. One commercial seed-washing procedure removed about 72% of the saponin contents. The initial saponin content (6.34%) reached a level as low as 0.25–0.01% during the first half hour of washing. Approximately 96% of saponin contents are removed from quinoa seed with washing for 60 min. The widely used methods are the conventional technologies of maceration, Soxhlet, and extraction using reflux.	[25,101]
GC-MS after silylation using N, O-Bis (trimethylsilyl) trifluoroacetamide (BSTFA). GC-MS/MS analysis to separate superfluous substances in the identification of saponin.	Hongcheon River Farming Union (Hongcheon, Korea)	Its accuracy, repeatability, and high linearity were appropriate for analyzing the saponins in quinoa with this method. The amounts of oleanolic acid, phytolaccagenic acid, and hederagenin were different among the different parts of the quinoa, including the sprouts and the fully grown quinoa plant parts. The saponin contents were highest in the quinoa seed bran and lowest in the quinoa leaves and roots.	[94]
Pressurized hot water extraction method (PHWE).	Andean plateau in Bolivia	There is a remarkable increase in saponin yield when the temperature exceeds 110 °C, with the highest amounts obtained at 195 °C (15.4 mg/g raw material).	[102]
Spectrophotometric analysis.	Provided by the INTA EEA Famailla’	The experimental extraction kinetics of the saponin contents from the quinoa seeds were studied at different water temperatures to improve the understanding of this process. From this study, the treatment carried out at 40 °C for 6 min can be considered the optimum one with which to reach a satisfactory level of saponins for human consumption without visible seed damage.	[103]
Gas chromatographic procedure.	Bio-Bio (Chile) Colorado (USA) Chile Maule (Chile)	Two-season trials support the low potential of the saponin contents for some of the selected quinoa accessions; however, this was strongly determined by the specific climatic conditions: higher saponins content in the rainy year and lower in the drier one. The large differences between the climatic conditions over the two seasons of the experimental trial allowed the assessment of plant behavior under drought stress.	[104]
Reversed-phase high-performance liquid chromatography (HPLC).	Central Chile	The experimental data were obtained through batch extraction with a ratio of quinoa to water of 1:10 under constant agitation, with a processing time between 15 and 120 min at 20, 30, 40, 50, and 60 °C. It was found that the residual saponin concentration in the quinoa seeds decreased as the washing temperature increased.	[105]

## Data Availability

Not applicable.

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
