# Peer review of "Challenges and Perspectives for Integrating Quinoa into the Agri-Food System"

_plants, 2023, doi:10.3390/plants12193361_

Round 1
Reviewer 1 Report
I recommend the publication of the article because the scientific experimentation on the highly nutritious and abiotic stress-tolerant crop quinoa that can be used to ensure the food security of the world's rapidly growing population under changing climatic conditions is of particular interest.
The aim and objectives of the article have been stated and are very interesting. The use of quinoa in food is an important topic, especially for the problems of world hunger and resistance to abiotic stresses. The work done is certainly of international interest and the format applied is certainly suitable for a research article. The work done is original, of particular interest and can certainly stimulate research on this topic. The length of the article is appropriate for the journal and the graphs and tables are clear and easy to understand. The conclusion summarises the aims of the work and future prospects.
Author Response
Dear Reviewer
Thank you for your valuable suggestions and comments for the improvement of the current manuscript. We have improved the manuscript based on your comments. All the mentioned changes have been incorporated in the manuscript. We appreciate your warm work earnestly and hope that the correction will meet with approval.

Reviewer 2 Report
1. What are the challenges considered by authors in their study?
2. What are the various limitations in your study?
3. Why Quinoa seeds considered in your research?
4. What is the impact of Quinoa seeds in your research?
5. Any effects due to water logging if yes state or justify
6. Give citations wherever required.
7. Compare your work with other researchers
8. Add latest literatures.
Author Response

(The authors gave the same response as above.)

Reviewer 3 Report
I have read this review paper. It is written in a very wide manner, however, many things are already available in the literature. Recently many review papers have been published in Plants-MDPI (https://www.mdpi.com/2073-4395/9/4/176 ; https://www.mdpi.com/2223-7747/12/4/868 ). I couldn't find much novelty in this review paper, hence, the authors need to be precise and up-to-date about their literature collection.
The title needs to be rewritten specifically.
Besides, the manuscript needs good polishing and the inclusion of the latest references.
In Table 1, many thanks are missing.
under the section, Biotic and abiotic stress tolerance, only two abiotic stresses are mentioned and no biotic stress is mentioned.
I have read this review paper. It is written in a very wide manner, however, many things are already available in the literature. Recently many review papers have been published in Plants-MDPI (https://www.mdpi.com/2073-4395/9/4/176 ; https://www.mdpi.com/2223-7747/12/4/868 ). I couldn't find much novelty in this review paper, hence, the authors need to be precise and up-to-date about their literature collection.
The title needs to be rewritten specifically.
Besides, the manuscript needs good polishing and the inclusion of the latest references.
In Table 1, many thanks are missing.
under the section, Biotic and abiotic stress tolerance, only two abiotic stresses are mentioned and no biotic stress is mentioned.
Author Response

(The authors gave the same response as above.)

Round 2
Reviewer 2 Report
Revision is appropriate
Reviewer 3 Report
The authors have now revised the manuscript, hence, I recommend it for publication in this journal.
It's fine